# Graph Policy Network for Transferable Active Learning on Graphs

**Shengding Hu[1], Zheng Xiong[1], Meng Qu[2,5], Xingdi Yuan[3], Marc-Alexandre Côté[3], Zhiyuan Liu[1], and Jian Tang[2,4,6]**

[1]Tsinghua University, [2]Mila-Quebec AI Institute, [3]Microsoft Research
[4]HEC Montreal, Canada, [5]Université de Montréal, [6]CIFAR AI Research Chair
hsd20@mails.tsinghua.edu.cn, xiongz17@tsinghua.org.cn, meng.qu@umontreal.ca,
{eric.yuan,macote}@microsoft.com, liuzy@tsinghua.edu.cn, jian.tang@hec.ca

## Abstract

Graph neural networks (GNNs) have been attracting increasing popularity due to their simplicity and effectiveness in a variety of fields. However, a large number of labeled data is generally required to train these networks, which could be very expensive to obtain in some domains. In this paper, we study active learning for GNNs, i.e., how to efficiently label the nodes on a graph to reduce the annotation cost of training GNNs. We formulate the problem as a sequential decision process on graphs and train a GNN-based policy network with reinforcement learning to learn the optimal query strategy. By jointly training on several source graphs with full labels, we learn a transferable active learning policy which can directly generalize to unlabeled target graphs. Experimental results on multiple datasets from different domains prove the effectiveness of the learned policy in promoting active learning performance in both settings of transferring between graphs in the same domain and across different domains.

## 1   Introduction

Graphs encode the relations between different objects and are ubiquitous in real world. Learning effective representation of graphs is critical to a variety of applications. Recently, graph neural networks (GNNs) have been attracting growing attention for their effectiveness in graph representation learning [30, 33]. They have achieved great success on various tasks such as node classification [15, 27] and link prediction [4, 32]. Despite their appealing performance, GNNs typically require a large amount of labeled data for training [29]. However, in many domains, such as chemistry [11] and health care [6], it could be very expensive and time-consuming to collect a large amount of labeled data, which significantly limits the performance of GNNs in these domains.

Active learning [1, 22, 25] is a promising strategy to tackle this challenge. The general idea of active learning is to dynamically query the labels of the most informative instances selected from the unlabeled data. Although active learning has been proven effective on independent and identically distributed (i.i.d.) data such as natural language processing [24] and computer vision [9], how to apply it to graph-structured data with dense interconnections between different instances remains under-explored. This motivates us to study active learning on graphs, i.e., how to efficiently label the nodes on a graph to reduce the annotation cost of training GNNs.

Towards this goal, several methods have been proposed recently [8, 13, 12, 3, 10, 5]. To measure the informativeness of each node, they either design a single selection criterion based on the graph characteristics, or adaptively combine several selection criteria. Then the most "informative" node is labeled at each query step according to the selection criterion. However, these methods suffer from the following limitations. (1) *Ignoring Long-term Performance*: Existing methods usually adopt

different selection criterion as a surrogate objective function and greedily optimize it at each query step. However, the long-term objective function we truly want to optimize is how to select a sequence of nodes which can maximize the final performance score of the GNN trained on them. Maximizing the short-term surrogate criterion usually leads to sub-optimal query strategies. (2) *Lack of Node Interactions*: When measuring node informativeness, existing methods usually consider each node independently and ignore the interconnections between different nodes. For example, if an unlabeled node has several labeled neighbors, then it is very likely that this node would provide little additional information for training.

To tackle these limitations, we propose GPA, a **G**raph **P**olicy network for transferable **A**ctive learning on graphs. Our approach formalizes active learning on graphs as a Markov decision process (MDP) and learns the optimal query strategy with reinforcement learning (RL), where the state is defined based on the current graph status, the action is to select a node for annotation at each query step, and the reward is the performance gain of the GNN trained with the selected nodes. By maximizing its long-term returns with policy gradient [26], our policy network can effectively learn to optimize the long-term performance of the GNN in an end-to-end fashion. Moreover, our approach parameterizes the policy network as another GNN to explicitly model node interactions, which effectively propagates useful information over the graph and thereby better measures node informativeness. We train the graph policy network on multiple training graphs where node labels are available, and directly transfer the learned policy to perform active learning on unseen test graphs where no labels are provided initially. The learned policy does not require any additional retraining or fine-tuning on the test graphs, thus is a zero-shot policy transfer.

We evaluate GPA on the standard semi-supervised node classification task under two experimental settings with increasing difficulty: 1) the training graphs and testing graphs are from the same domain; 2) the training graphs and testing graphs are from different domains. Experimental results prove the effectiveness of GPA over competitive baselines under both settings.

## 2  Related work

**Graph Neural Networks**. Typically, GNNs [30, 33] learn node representation by iteratively aggregating neighborhood information of each node. For example, GCN [15] aggregates neighborhood information using a convolution operator, while GAT [27] utilizes an attention mechanism in the aggregation process. Despite their effectiveness, GNNs typically require a large number of labeled data for training, which entails high annotation cost in some domains [16]. Consequently, we propose to study active learning on graphs to reduce the annotation cost of training GNNs.

**Active Learning on graphs**. Active learning [1, 25, 22, 2] has been widely studied on i.i.d. data in different domains such as natural language processing [24] and computer vision [9]. Recently, there are also a handful of studies focusing on active learning for graph-structured data. Some earlier studies [8, 12, 13] are developed based on the graph homophily assumption that neighboring nodes are more likely to have the same label, and utilize theories of graph signal processing to select nodes for active learning. Compared to these earlier studies, our proposed method leverages recent advancements in RL and GNNs to better measure node informativeness. Moreover, our method doesn't rely on the strict homophily assumption, thus can be applied to a larger variety of graphs in the real world. More recent works utilize the expressive power of GNNs to design more informative selection criteria. AGE [3] measures node informativeness by a linear combination of three heuristics, where the combination weights are sampled from a beta distribution with time-sensitive parameters. ANRMAB [10] also uses the combination of different heuristics, but dynamically adjusts the combination weights based on a multi-armed bandit framework. Similarly, ActiveHNE [5] tackles active learning on heterogeneous graphs by posing it as a multi-armed bandit problem. However, all these methods measure the informativeness of different nodes independently, without explicitly considering their interactions. Moreover, these methods either greedily choose the most informative node in each single query step (AGE) or maximize a surrogate reward signal (ANRMAB and ActiveHNE), but fail to directly optimize the long-term performance gain of the whole query sequence.

**Reinforcement Learning (RL) for Active Learning**. There are also some studies trying to approach active learning through reinforcement learning. Fang et al. [7] uses deep Q-networks (DQN) [19] to learn active learning policies for named entity recognition (NER). Similarly, Liu et al. [17] uses imitation learning to select the most informative data points for NER. Liu et al. [18] tackles active learning for neural machine translation with reinforcement learning. However, all these studies

focus on i.i.d. data. In contrast, we focus on graph-structured data, where different nodes are highly correlated. Specifically, our approach learns a GNN-based policy network to utilize the interactions between different nodes.

# 3 Methodology

## 3.1 Problem definition

We consider a graph denoted as $G = (V, E)$, where $V$ is a set of nodes and $E$ is a set of edges. Each node $v \in V$ is associated with a feature vector $x_v \in \mathcal{X} \subseteq \mathbb{R}^d$, and a label $y_v \in \mathcal{Y} = \{1, 2, \cdots, C\}$. The node set is divided into three subsets as $V_{\text{train}}$, $V_{\text{valid}}$, and $V_{\text{test}}$. In conventional semi-supervised node classification, the labels of a subset $V_{\text{label}} \subseteq V_{\text{train}}$ are given. The task is to learn a classification network $f_{G, V_{\text{label}}}$ (formulated as a GNN) with the graph $G$ and $V_{\text{label}}$ to classify the nodes in $V_{\text{test}}$.

For active learning on graphs, the labeled training subset is initialized as an empty set, $V^0_{\text{label}} = \emptyset$. A query budget $B$ is given, which allows us to sequentially acquire the labels of $B$ samples from $V_{\text{train}}$, where $B \ll |V_{\text{train}}|$. At each step $t$, we select an unlabeled node $v_t$ from $V_{\text{train}} \backslash V^{t-1}_{\text{label}}$ based on an active learning policy $\pi$ and query the label of $v_t$. Next, we update the labeled node set as $V^t_{\text{label}} = V^{t-1}_{\text{label}} \cup \{v_t\}$. The classification GNN $f$ is then trained with the updated $V^t_{\text{label}}$ for one more epoch. When the budget $B$ is used up, we stop the query process and continue training the classification GNN $f$ with $V^B_{\text{label}}$ until convergence.

We learn the active learning policy $\pi$ on several labeled training graphs with reinforcement learning, and evaluate the learned policy on initially unlabeled test graphs without fine-tuning or retraining the policy, i.e., zero-shot policy transfer. This zero-shot policy transfer setting is essential for our method, as our goal is to perform active learning on unlabeled test graphs, thus we should not acquire any additional labels beyond the query budget on these graphs to tune the policy.

During the training phase (Section 3.4), we collect a set of source graphs $\mathcal{G}_S$ with node labels, and learn the optimal policy $\pi^*$ to maximize $\sum_{G \in \mathcal{G}_S} \mathcal{M}(f_{G, V^B_{\text{label}}})$, where $\mathcal{M}(\cdot)$ is a metric used to evaluate the performance of the classification GNN; $V^B_{\text{label}} = (v_1, \ldots, v_B)$ is the node sequence labeled by $\pi^*$ on graph $G$ under the annotation budget $B$. During the evaluation phase (Section 3.5), we directly apply the learned policy $\pi^*$ to a set of target graphs $\mathcal{G}_T$. For each $G \in \mathcal{G}_T$, we have no node labels initially, then we select a sequence of nodes to label based on $\pi^*$ and use them to train the classification GNN on $G$. Our ultimate goal is to learn the transferable active learning policy $\pi^*$ from $\mathcal{G}_S$ which can perform well on $\mathcal{G}_T$ without fine-tuning or retraining (zero-shot policy transfer).

## 3.2 Active learning on graphs as MDP

In the task of active learning for GNNs, our goal is to interactively select a sequence of nodes which maximize the performance of the GNN trained on them. This problem could be naturally formalized as an MDP. Intuitively, given the condition of the current graph (i.e., predictions of the classification network and information about available labels in the graph) as state, the active learning system takes an action by selecting the next node to query. It is then rewarded by the performance gain of the classification GNN trained with the updated set of labeled nodes. Formally, the MDP is defined as follows.

**State**. We denote the state in graph $G$ at step $t$ as a matrix $S^t_G$, where each row $\mathbf{s}^t_v$ is the state representation of node $v$. We define the state representation of each node based on several commonly-used heuristic criteria in active learning [22] as follows.

- We compute the degree of a node to measure its representativeness. The intuition is that high-degree nodes are likely to be hubs in a graph, and thus their labels are likely more informative. To ensure computational stability, we scale node degree by a hyperparameter $\alpha$ and clip it to 1, i.e.,

$$\mathbf{s}^t_v(1) = \min(\text{degree}(v)/\alpha, 1).$$

- We compute the entropy of the label distribution predicted by the classification GNN $f$ on each node to measure its uncertainty. If the network is not confident about its prediction on certain nodes, then the labels of these nodes are likely more useful. We divide the entropy $H$ by $\log(\#\text{classes})$ to fit its value range within $[0, 1]$, which ensures its transferability across graphs even with different

class numbers, i.e.,

$$\mathbf{s}_v^t(2) = H(\bar{y}(v;t))/\log(\#\text{classes}),$$

where $\bar{y}(v;t) \in \mathbb{R}^C$ is the class probability of node $v$ predicted by the classification GNN at step $t$.

- In addition to the entropy of the node itself, we are also interested in the divergence between a node's predicted label distribution and its neighbor's. The divergence between neighboring nodes measures local graph similarity, which can help the active learning policy better identify potential clusters and decision boundaries in the graph. Consequently, we compute the average KL divergence and reverse KL divergence between the predicted label distribution of a node $v$ and its neighbors $N_v$ as a measure of local similarity, i.e.,

$$\mathbf{s}_v^t(3) = \frac{1}{|N_v|} \sum_{u \in N_v} \text{KL}(\bar{y}(v;t) \,||\, \bar{y}(u;t)), \ \mathbf{s}_v^t(4) = \frac{1}{|N_v|} \sum_{u \in N_v} \text{KL}(\bar{y}(u;t) \,||\, \bar{y}(v;t)).$$

- We use an indicator variable to represent whether a node has been labeled or not, i.e.,

$$\mathbf{s}_v^t(5) = \mathbb{1}\{v \in V_{\text{label}}^{t-1}\}.$$

We concatenate these features to form the state representation of each node $s_v^t$. In addition, we can easily incorporate additional heuristic features (e.g., the features proposed in [3, 10, 5]) into this flexible framework by appending them to the node representation vector. In this paper, however, we only use the five primary features introduced above and utilize the policy network to automatically learn more complicated and informative criterion for node selection. The graph state matrix $S_G^t$ will be passed into the policy network $\pi$ to generate action probabilities.

**Action**. At time step $t$, the action is to select a node $v_t$ from $V_{\text{train}} \backslash V_{\text{label}}^{t-1}$ based on $p_G^t \sim \pi(\cdot|S_G^t)$, the action probability given by the policy network in step $t$. (See Section 3.3 for the calculation of $p_G^t$ )

**Reward**. We use the classification GNN's performance score on the validation set after convergence as a trajectory reward. Although this trajectory reward is delayed and sparse compared to using step-wise performance gain as immediate rewards, empirically it provides a much more stable estimation of the policy's quality, as it is more robust to the random interference factors during the training process of the classification GNN. Given a sequence of labeled nodes $V_{\text{label}}^B = (v_1, \ldots, v_B)$, we define the trajectory reward with respect to $V_{\text{label}}^B$ as

$$R(V_{\text{label}}^B) = \mathcal{M}(f_{G,V_{\text{label}}^B}(V_{\text{valid}}), y_{\text{valid}}), \tag{1}$$

where $f_{G,V_{\text{label}}^B}$ is the classification GNN trained with the graph $G$ and the labels of $V_{\text{label}}^B$; $V_{\text{valid}}$ and $y_{\text{valid}}$ are the nodes and labels of the validation set; $\mathcal{M}$ is the evaluation metric.

**State Transition Dynamics**. At each query step $t$, a newly labeled node $v_t$ is added to $V_{\text{label}}^t$ to update the classification GNN, the graph state thus transits from $S_G^t$ to $S_G^{t+1}$. Specifically, the selection indicator in the state vector of the selected node $v_t$ is changed from 0 to 1. Updating the classification GNN can influence the predicted label distribution, thus the entropy and KL divergence terms in the state vector of each node will change accordingly. Since it is hard to directly model the transition dynamics $p(S_G^{t+1}|S_G^t, v_t)$, we learn the optimal policy in a model-free approach.

**Framework**. We show an overview of the policy training framework in Figure 1. At query step $t$, we first update the current graph state $S_G^t$ with the graph $G$ and the outputs of the classification GNN $f_{G,V_{\text{label}}^{t-1}}$. The policy network $\pi$ takes $S_G^t$ as input and produces a probability distribution over actions as $p_G^t$, which represents the probability of annotating each unlabeled node in the candidate pool $V_{\text{train}} \backslash V_{\text{label}}^{t-1}$. Next, we sample a node $v_t$ based on $p_G^t$ for annotation and add it to the labeled training subset to get $V_{\text{label}}^t$. The classification GNN $f$ is trained for one more epoch with $V_{\text{label}}^t$ to get $f_{G,V_{\text{label}}^t}$, which is then used to generate the graph state $S_G^{t+1}$ for the next step. When $t = B$, we stop the query phase and train $f$ until convergence. Finally, we evaluate $f_{G,V_{\text{label}}^B}$ on the validation set $V_{\text{valid}}$, and the performance score is used as the trajectory reward $R$ to update the policy network $\pi$.

### 3.3 Policy network architecture

A key characteristic of active learning on graphs is that the nodes are highly correlated with each other based on the graph topology. This provides valuable information on the informativeness of each candidate node at different query step. To automatically extract such information and model

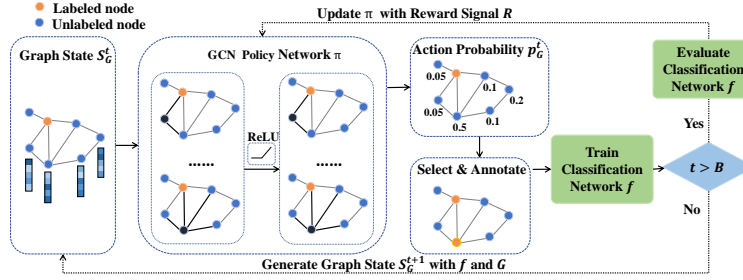

Figure 1: The RL-based framework for active learning on GNNs. Blue and orange nodes represent unlabeled and labeled nodes in the training set respectively. The navy blue nodes and bold edges show the information aggregating process of the policy network. For simplicity, we omit the validation nodes and test nodes. In the policy network $\pi$, each column represents a layer of GNN, and the graphs in each column correspond to the feature aggregation on different nodes.

the influence of graph structures on the query policy, we parameterize the policy network $\pi$ as a GNN, which iteratively aggregates neighborhood information to update the state representation of each node. Specifically, we implement the policy network as a $L$-layer Graph Convolutional Network (GCN) [15]. The propagation rule for layer $l$ in GCN is

$$H^{(l+1)} = \sigma\left(\tilde{D}^{-\frac{1}{2}}\tilde{A}\tilde{D}^{-\frac{1}{2}}H^{(l)}W^{(l)}\right), \tag{2}$$

where $W^{(l)}$ and $H^{(l)}$ are the weight and the input feature matrices of layer $l$ respectively. $\tilde{A} = A + I$ is the adjacency matrix with self loops, $\tilde{D}$ is a diagonal matrix with $\tilde{D}_{ii} = \sum_i \tilde{A}_{ij}$. We use ReLU as the activation function $\sigma$. In the first layer, we use $H^{(0)} = S_G^t$ as the initial input feature.

On top of the GCN, we apply a linear layer (with output dimension of 1) to the final output embedding $H^{(L)}$, and get a real number score for each node. These scores are then normalized by a softmax to generate a probability distribution $p_G^t$ over all candidate nodes $V_{\text{train}}/V_{\text{label}}^{t-1}$ for annotation in $G$:

$$p_G^t = \pi(\cdot|S_G^t) = \text{Softmax}(WH^{(L)} + b). \tag{3}$$

### 3.4 Policy training

In the training phase, given a set of $N_S$ labeled source graphs $\mathcal{G}_S = \{G_i | i = 1, \dots, N_S\}$, our goal is to maximize the sum of expected rewards obtained from following policy $\pi$ over the training graphs. The objective function with respect to the policy network parameters $\theta$ is

$$J(\theta) = \sum_{i=1}^{N_S} \mathbb{E}_{P(V_{\text{label}}^{B_i} = (v_1, \dots, v_{B_i}); \theta)} [R_i(V_{\text{label}}^{B_i})], \tag{4}$$

where $B_i$ is the query budget on graph $G_i$, and $R_i$ is the trajectory reward on graph $G_i$. We utilize REINFORCE [28], a classical policy gradient method, to train the policy network. For joint training on multiple source graphs, we iterate over all the training graphs in each episode to update the policy network. As different query sequences will be tried during policy training, we require full label information on these training graphs.

### 3.5 Policy transfer and evaluation

In the evaluation phase, given a set of $N_T$ unlabeled target graphs $\mathcal{G}_T = \{G_i | i = 1, \dots, N_T\}$, we directly apply the learned policy $\pi_\theta$ on each test graph to perform active learning.

Specifically, on each test graph $G \in \mathcal{G}_T$, we start with no labeled nodes initially. In each query step $t$, we first compute the state vector of each node and get the state matrix $S_G^t$ of graph $G^1$. Then the state matrix is passed into the policy network $\pi_\theta$ to compute the action probability, where the policy parameters $\theta = \{W^{(0)}, \dots, W^{(L)}, W, b\}$ are directly copied from the training phase[2], while the graph

topology matrices $\tilde{D}, \tilde{A}$ are substituted with the ones of the test graph. We then choose the candidate node with the maximal action probability to label and add it to the labeled training set to update the classification network. The predictions of the updated classification network are then used to generate the state matrix for the next query step. We repeat the above process until the query budget is reached, and finally train the classification network until convergence. Due to the space limit, we give the detailed pseudo-code for policy training and transfer in Appendix A.

## 4 Experiments and analysis

### 4.1 Experimental setup

**Datasets.** For transferable active learning on graphs from the *same* domain, we use a multi-graph dataset collected from Reddit[3], which consists of 5 graphs. For transferable active learning on graphs from *different* domains, we adopt 5 widely used benchmark datasets: Cora, Citeseer and Pubmed, Coauthor-Physics and Coauthor-CS [23]. We also use the 5 Reddit graphs in this setting, resulting 10 graphs in total. Due to the space limit, we give detailed description of these datasets in Appendix B.

**Baselines.** We compare our method against the following baseline methods:
**(1) Random:** At each step, randomly select a node to annotate. This is equivalent to the conventional semi-supervised training of GNNs.
**(2) Uncertainty-based policy:** At each step, predict the label distribution of each node with the current classification GNN, then annotate the node with the maximal entropy on label distribution.
**(3) Centrality-based policy:** At each step, annotate the node with the largest degree.
**(4) Coreset [21]:** Coreset performs k-means clustering over the outputs of the last hidden layer of the classification network, which was originally proposed for Convolutional Neural Networks. We simply apply this method on the node representation learned by the classification GNN. At each step, the node which is closest to the cluster center is selected for annotation.
**(5) AGE [3]:** AGE measures the informativeness of each node by combining three heuristics, i.e., the entropy of the predicted label distribution, the node centrality score, and the distance between the node's embedding and it nearest cluster center. To apply AGE under the transfer learning setting, we find the optimal combination weights on each training graph separately using grid search, then use their mean value as the combination weights on test graphs.
**(6) ANRMAB [10]:** ANRMAB utilizes the same set of selection heuristics as in AGE, and proposes a multi-armed bandit framework to dynamically adjust the combination weights of these heuristics. ANRMAB uses the performance score on historical query steps as the rewards to the multi-armed bandit machine, which enables it to learn the combination weights during the query process.[4]

**Evaluation Metrics and Parameter Settings.** Following the common settings in GNN literature [31], we use Micro-$F_1$ and Macro-$F_1$ as the evaluation metrics. On each graph, we set the sizes of validation and test sets as 500 and 1000 respectively and use all remaining nodes as the candidate training samples for annotation. To test the policy, we run 100 independent experiments with different classification network initialization and report the average performance score on the test set.

We implement the policy network as a two-layer GCN [15] with a hidden layer size of 8. We use Adam [14] as the optimizer with a learning rate of 0.01. The policy network is trained for a maximum of 2000 episodes with a batch size of 5. To demonstrate the advantage of active learning on reducing annotation cost, we set the query budget on each graph as ($5 \times$ #classes), which is far less than the default labeling budget of ($20 \times$ #classes) in conventional (semi-supervised) GNN literature [15]. We set the scaling hyperparameter as $\alpha = 20$ based on the average graph degree in our datasets. We use Micro-$F_1$ as the reward metric. For the classification network, we implement it as a two-layer GCN with a hidden layer size of 64. We use Adam as the optimizer with a learning rate of 0.03 and a weight decay of 0.0005. [5]

Table 1: Results of transferable active learning on graphs from the *same* domain. The active learning policy is trained on Reddit {1, 2} and evaluated on Reddit {3, 4, 5}. **Boldface** and underline represent the best and second best scores respectively.

| Method | Reddit3 | | Reddit4 | | Reddit5 | |
|---|---|---|---|---|---|---|
| | Micro-$F_1$ | Macro-$F_1$ | Micro-$F_1$ | Macro-$F_1$ | Micro-$F_1$ | Macro-$F_1$ |
| Random | 88.21 | 87.30 | 84.81 | 79.87 | 86.20 | 84.39 |
| Uncertainty | 70.03 | 64.34 | 72.28 | 60.32 | 73.27 | 63.67 |
| Centrality | 90.93 | 90.35 | 83.43 | 75.96 | 84.83 | 79.71 |
| Coreset | 78.34 | 76.11 | 82.18 | 76.71 | 83.29 | 81.99 |
| AGE | 91.09 | 90.44 | 87.55 | 84.39 | 88.02 | 85.99 |
| ANRMAB | 85.26 | 83.06 | 83.14 | 76.80 | 83.65 | 79.99 |
| GPA (Ours) | **92.85** | **92.53** | **91.57** | **89.46** | **91.60** | **91.38** |

Table 2: Results of transferable active learning on graphs from *different* domains. Cora and Citeseer are used as the training graphs, while all other graphs are used for evaluation. For simplicity, we discard the three single-heuristic methods due to their relatively low performance.

| Method | Metric | Pubmed | Reddit1 | Reddit2 | Reddit3 | Reddit4 | Reddit5 | Physics | CS |
|---|---|---|---|---|---|---|---|---|---|
| Random | Micro-$F_1$ | 68.35 | 81.88 | 91.19 | 87.76 | 85.37 | 86.45 | 82.03 | 84.70 |
| | Macro-$F_1$ | 67.57 | 80.26 | 89.92 | 86.12 | 80.89 | 84.52 | 70.77 | 70.57 |
| AGE | Micro-$F_1$ | 74.78 | 83.76 | 92.56 | 90.61 | 86.94 | 87.73 | 84.68 | 86.33 |
| | Macro-$F_1$ | 73.26 | 82.81 | 91.61 | 89.99 | 83.15 | 85.88 | 77.25 | 80.63 |
| ANRMAB | Micro-$F_1$ | 69.35 | 81.25 | 88.74 | 85.26 | 83.14 | 83.65 | 82.55 | 86.63 |
| | Macro-$F_1$ | 68.68 | 79.43 | 86.58 | 83.06 | 76.80 | 79.99 | 71.57 | 79.00 |
| GPA (Ours) | Micro-$F_1$ | **77.80** | **88.10** | **95.19** | **92.07** | **91.39** | **90.66** | **88.08** | **87.90** |
| | Macro-$F_1$ | **75.66** | **87.75** | **95.00** | **91.77** | **89.60** | **90.22** | **82.82** | **84.99** |

## 4.2 Transferable active learning on graphs from the same domain

Table 1 shows the results of transferable active learning on graphs from the same domain. Our policy successfully transfers to all the three test graphs of Reddit under a zero-shot transfer setting.

Surprisingly, for the three methods which use a single heuristic as the selection criterion, i.e., *uncertainty*, *centrality* and *Coreset*, none of them could consistently outperform random selection. We suspect the reason is that the distribution of the selected samples is different from the underlying distribution of all the nodes. For example, when using *uncertainty* for selection, nodes that are close to the decision boundaries usually have higher entropy and are more likely to be selected, which introduces a distribution drift to the labeled training data. Consequently, a combination of different heuristics is essential to effectively measure node informativeness — as is done in our method.

Compared to the two baselines that combine different heuristics as the selection criterion, GPA consistently outperforms AGE and ANRMAB. The reasons of the performance gain are twofold. First, our approach formulates the active learning problem as an MDP, which directly optimizes the long-term performance of the classification GNN. Second, our approach parameterizes the policy network as a GNN, which could leverage node interactions to better measure node informativeness. The underlying reason for ANRMAB's inferior performance to AGE may be that for ANRMAB, to learn good weights of different heuristics, at least a moderate number of labeled data are required. In our setting, we focus on very limited query budgets, which makes learning good weights more difficult.

## 4.3 Transferable active learning on graphs across different domains

Table 2 shows the results of transferable active learning on graphs across different domains. Our policy successfully transfers to graphs across different domains and achieves the best performance on all the test graphs. Furthermore, comparing with Table 1, we can see that the cross-domain GPA policy performs comparably to the single-domain GPA policy on Reddit {3, 4, 5}, which again suggests the strong transferability of our approach. This is mainly because that our policy is learned on a state space closely related to the classification GNN's learning process, instead of depending on any graph-specific features. Moreover, the policy network is optimized jointly over multiple graphs, which helps it learn an universal policy that is naturally transferable to different graphs. Due to the space limit, please refer to Appendix C.1 for further experiments on different training graphs.

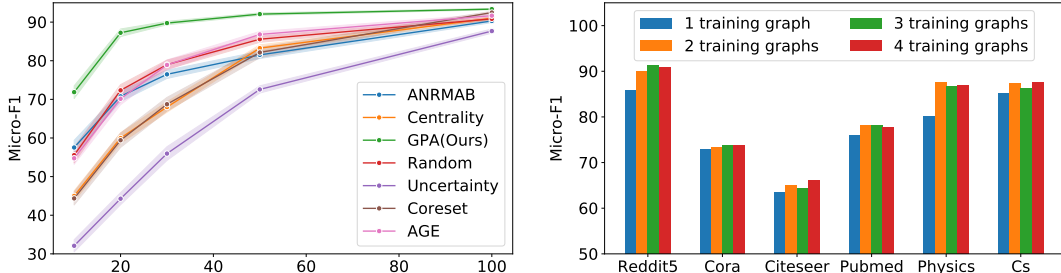

Figure 2: Left: Performance of different methods on Reddit 4 under different query budgets. The x-axis represents label budget, and the y-axis represents the Micro-$F_1$ score. Right: Performance of GPA on different test graphs when trained with different number of source graphs.

Table 3: Performance comparison between using GCN and MLP as the policy network.

| Method | Metric | Pubmed | Reddit1 | Reddit2 | Reddit3 | Reddit4 | Reddit5 | Physics | CS |
|---|---|---|---|---|---|---|---|---|---|
| GCN | Micro-$F_1$ | **77.80** | **88.10** | **95.19** | **92.07** | **91.39** | **90.66** | **88.08** | 87.90 |
| | Macro-$F_1$ | **75.66** | **87.75** | **95.00** | **91.77** | **89.60** | **90.22** | **82.82** | 84.99 |
| MLP | Micro-$F_1$ | 70.49 | 72.65 | 85.61 | 81.66 | 80.03 | 81.16 | 80.30 | **88.53** |
| | Macro-$F_1$ | 70.08 | 67.56 | 82.11 | 77.63 | 70.99 | 71.43 | 68.86 | **85.91** |

## 4.4 Performance under different query budgets

Next, we compare all the algorithms on the dimension of query budgets. In this study, Reddit is used as an example. We train our policy on Reddit $\{1, 2\}$ with $\{10, 20, 30, 50, 100\}$ budgets, then evaluate the learned policy on Reddit 4 under the corresponding budgets. All baseline methods are also tested using the same set of budgets. We test each method under each budget for 100 times and report the averaged Micro-$F_1$ score with 95% confidence interval. Figure 2 (left) shows that our policy consistently outperforms all baselines under all budgets. Compared with random selection, which uses 100 budget to reach a Micro-$F_1$ of 90.0, our approach only needs 30 budget to reach the same result. Meanwhile, AGE uses 100 budget to reach a Micro-$F_1$ of 91.7, while our approach only uses 50 budget to achieve the same result. We also notice that using only half of the full budget (50), GPA can already achieve a higher Micro-$F_1$ than most of the baselines consuming 100 budget. Due to the space limit, please refer to Appendix C.2 for further experiments on how different query budgets influence the performance of the active learning policy.

## 4.5 Ablation study

**Number of Training Graphs**    We study the performance and transferability of the learned policy w.r.t. the number of training graphs. We select $\{1, 2, 3, 4\}$ graphs from Reddit as the training graphs, and evaluate on the remaining 6 graphs. The result is shown in Figure 2 (right). On average, the policy trained on multiple graphs transfers better than the policy trained on a single graph. The main reason may be that training on a single graph overfits to the specific pattern of the training graph, while training on multiple graphs better captures the general pattern across different graphs.

**Importance of Modeling Node Interactions**    Our GNN-based policy network models node informativeness by considering graph structures. Here, we compare it with a policy network without taking graph structures into consideration. We parameterize the policy network as a multi-layer perceptron (MLP), which only utilizes single-node information. For a fair comparison, we use a 3-layer MLP with a hidden layer size of 8 — it has the same number of parameters as the GCN policy network. We train the two policy networks on Cora + Citeseer, and evaluate on the remaining graphs. As shown in Table 3, GCN outperforms MLP by a large margin on all test graphs except Coauthor-CS, which evinces the importance of modeling node interactions.

**Contribution of State Features**    We also investigate the contribution of the state features introduced in Section 3.2. We remove each of them from the state space to see how they influence the learned policy. We take the Reddit dataset as an example. As shown in Table 4, removing any of the features can result in a performance drop, which validates the effectiveness of these features. Among the four features, the binary label indicator seems to contribute the most to the policy's performance. We believe the reason is that propagating the annotation information over the graph helps the policy better identify and model the under-explored areas in the graph.

Table 4: Contribution of each state feature. The policies are trained on Reddit {1, 2} and evaluated on Reddit {3, 4, 5}. Each row corresponds to removing one feature, while "GPA (full model)" means using all the features.

| Features | Reddit3 | | Reddit4 | | Reddit5 | |
|---|---|---|---|---|---|---|
| | Micro-$F_1$ | Macro-$F_1$ | Micro-$F_1$ | Macro-$F_1$ | Micro-$F_1$ | Macro-$F_1$ |
| GPA (full model) | **92.85** | **92.53** | **91.57** | 89.46 | **91.60** | **91.38** |
| − Entropy | 92.48 | 92.12 | 90.12 | 86.94 | 90.35 | 89.88 |
| − Degree | 92.36 | 92.07 | 91.48 | **89.55** | 90.56 | 90.16 |
| − KL | 92.76 | 92.47 | 91.12 | 88.23 | 91.29 | 90.98 |
| − Indicator | 91.11 | 90.20 | 88.90 | 86.12 | 90.72 | 90.41 |

## 4.6 Case study

To better understand how the learned policy works, we conduct a case study by visualizing the node sequence selected by different method on a toy graph collected from Reddit ($|V| = 85, |E| = 156, |\mathcal{Y}| = 4$). We apply the cross-domain policy learned in Section 4.3 to this toy graph for 100 times with different classification network initialization, and report the most frequently selected node in each query step. For comparison, we also report the node sequence selected by AGE and the centrality criterion. Since ANRMAB needs to learn during the query process, it fails to perform meaningful selection on this toy graph with small size, and thus is not reported.

As shown in Figure 3, GPA better explores the whole graph space compared to the others two methods, which helps it to better model the data distribution and thus achieves much better performance. Specifically, the node sequences selected by AGE and the centrality criterion are very similar, both biased towards the nodes with large degree. On the contrary, GPA not only utilizes the nodes with large degree (e.g. node 1, 2), but also fully explores the under-represented areas (e.g. node 10, 13, 15) in the graph based on its annotation trajectory. It also learns to automatically switch between different classes to reach a class balance (e.g. no two consecutively selected nodes belong to the same class).

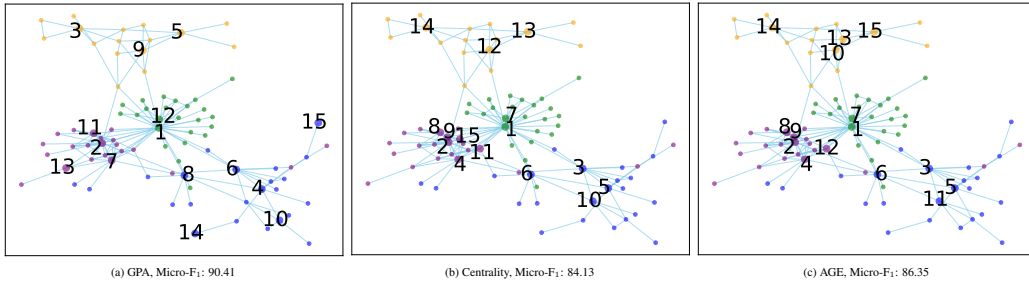

(a) GPA, Micro-$F_1$: 90.41      (b) Centrality, Micro-$F_1$: 84.13      (c) AGE, Micro-$F_1$: 86.35

Figure 3: Visualization of the node query process on a toy graph from Reddit. The query budget is 15. Each color represents a unique class. The annotated nodes are magnified, and the numbers represent at which step they are selected. The Micro-$F_1$ score of each strategy is reported in the subfigure caption. For comparison, random selection achieves a Micro-$F_1$ score of 84.74.

## 5 Conclusion

In this paper, we investigate active learning on graphs which aims to reduce the annotation cost for training GNNs. We formulate the problem as an MDP and learn a transferable query policy with RL. We parameterize the policy network as a GNN to explicitly model graph structures and node interactions. Experimental results suggest that our approach can transfer between graphs in both single-domain and cross-domain settings, substantially outperforming competitive baselines. For future work, we consider to dynamically adjust the importance of different state features based on the current time step [3, 10], or to incorporate global graph information into the policy. We can also try applying our framework to other data types by modeling the correlations between the data samples as a graph.

## Broader Impact

Graph-structured data are ubiquitous in real world, covering a variety of domains and applications such as social science, biology, medicine, and political science. In many domains such as biology and medicine, annotating a large number of labeled data could be extremely expensive and time consuming. Therefore, the algorithm proposed in this paper could help significantly reduce the labeling efforts in these domains — we can train systems on domains where labeled data are available, then transfer to those lower-resource domains.

We believe such systems can help accelerating some research and develop processes that usually take a long time, in domains such as drug development. It can potentially also lower the cost for such research by reducing the need of expert-annotations.

However, we also acknowledge potential social and ethical issues related to our work.

1. Our proposed system can effectively reduce the need of human annotations. However, in a broader point of view, this can potentially lead to a reduction of employment opportunities which may cause layoff to data annotators.

2. GNNs are widely used in domains related to critical needs such as healthcare and drug development. The community needs to be extra cautious and rigorous since any mistake may cause harm to patients.

3. Training the policy network for active learning on multiple graphs is relatively time - and computational resource - consuming. This line of research may produce more carbon footprint compared to some other work. Therefore, how to accelerate the training process by developing more efficient algorithms requires further investigation.

Nonetheless, we believe that the directions of active learning and transfer learning provide a hopeful path towards our ultimate goal of data efficiency and interpretable machine learning.

## Acknowledgments and Disclosure of Funding

This project is supported by the Natural Sciences and Engineering Research Council (NSERC) Discovery Grant, the Canada CIFAR AI Chair Program, collaboration grants between Microsoft Research and Mila, Amazon Faculty Research Award, Tencent AI Lab Rhino-Bird Gift Fund and a NRC Collaborative R&D Project (AI4D-CORE-06).

## Footnotes

[1]Since all the state features defined in Section 3.2 can be naturally transferred across graphs even with different class numbers and graph topology, the test graph shares a consistent state space with the training graphs.

[2]Note that all the weight matrices in the policy network are node-wise operators which operate on the feature dimension (i.e., the second dimension) of the state matrix. Thus as long as different graphs share the same state space, the same policy network can be naturally reused across graphs with different node numbers.

[3]Reddit is an online forum where users create posts and comment on them.

[4]As the code of ANRMAB is not provided, we implement it based on the pseudo-code from their paper.

[5]Our code is publicly available at https://github.com/ShengdingHu/GraphPolicyNetworkActiveLearning

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
