[Supplementary Material · Appendix.pdf]

# A  Algorithm details

In this section, we present the pseudo-code of our approach for both policy training (Algorithm 1) and evaluation (Algorithm 2).

---

**Algorithm 1:** Train the policy on multiple labeled source graphs

---

**Input:** the labeled training graphs $\mathcal{G}_T = \{G_i\}$, the query budget on each graph $\{B_i\}$, the maximal training episode $M$

**Result:** the query policy $\pi_\theta$

Randomly initialize $\pi_\theta$;

**for** *episode = 1 to M* **do**
    **for** *$G_i$ in $\mathcal{G}_T$* **do**
        $V_{\text{label}}^0 \leftarrow \emptyset$;
        Randomly initialize the classification GNN as $f_{G_i, V_{\text{label}}^0}$;
        **for** *$t = 1$ to $B_i$* **do**
            Generate the graph state $S_{G_i}^t$ based on $f_{G_i, V_{\text{label}}^{t-1}}$ and $G_i$;
            Compute the probability distribution over candidate nodes as $p_{G_i}^t = \pi_\theta(S_{G_i}^t)$;
            Sample an unlabeled node $v_t \sim p_{G_i}^t$ from $V_{\text{train}} \backslash V_{\text{label}}^{t-1}$ and query for its label;
            $V_{\text{label}}^t \leftarrow V_{\text{label}}^{t-1} \cup \{v_t\}$;
            Train the classification GNN for one more epoch with $V_{\text{label}}^t$ to get $f_{G_i, V_{\text{label}}^t}$;
        **end**
        Train the classification GNN $f_{G_i, V_{\text{label}}^{B_i}}$ until converge;
        Evaluate $f_{G_i, V_{\text{label}}^{B_i}}$ on the validation set to get the reward signal $R(V_{\text{label}}^{B_i})$;
        Use $R(V_{\text{label}}^{B_i})$ to update $\pi_\theta$ with policy gradient;
    **end**
**end**

---

---

**Algorithm 2:** Evaluate the learned policy on an unlabeled test graph

---

**Input:** the unlabeled test graph $G$, the learned policy $\pi_\theta$, the query budget $B$ on graph $G$

**Result:** the classification GNN $f$ trained on the selected node sequence $\tau$

$\tau = [\,]$;

$V_{\text{label}}^0 \leftarrow \emptyset$;

Randomly initialize the classification GNN as $f_{G, V_{\text{label}}^0}$;

**for** *$t = 1$ to $B$* **do**
    Generate the graph state $S_G^t$ based on $f_{G, V_{\text{label}}^{t-1}}$ and $G$;
    Compute the probability distribution over candidate nodes as $p_G^t = \pi_\theta(S_G^t)$;
    Select $v_t = \arg\max_v p_G^t(v)$ from $V_{\text{train}} \backslash V_{\text{label}}^{t-1}$ and query for its label;
    $V_{\text{label}}^t \leftarrow V_{\text{label}}^{t-1} \cup \{v_t\}$;
    Train the classification GNN for one more epoch to get $f_{G, V_{\text{label}}^t}$;
    $\tau$.append($v_t$);
**end**
Train $f_{G, V_{\text{label}}^B}$ until converge;
Evaluate the converged classification GNN $f$ on the test set $V_{\text{test}}$ of $G$.

---

# B  Dataset descriptions

Here we present the details of the datasets used in our experiments.

Table 5: Statistics of the datasets used in our experiments. For Reddit, * represents the average value over all individual graphs. The Budget column shows the query budget on each graph, which is set to $5 \times$ #class by default. We use Physics and CS as the abbreviation of Coauthor-Physics and Coauthor-CS respectively.

| Dataset | Nodes | Edges | Features | Classes | Budget |
|---------|-------|-------|----------|---------|--------|
| Cora | 2708 | 5278 | 1433 | 7 | 35 |
| Citseer | 3327 | 4676 | 3703 | 6 | 30 |
| Pubmed | 19718 | 44327 | 500 | 3 | 15 |
| Physics | 34493 | 247962 | 2000 | 5 | 25 |
| Cs | 18333 | 81894 | 6805 | 15 | 75 |
| Reddit | 4017.6* | 28697.6* | 300 | 10 | 50 |

For transferable active learning on graphs from the same domain, we use a multi-graph dataset collected from Reddit[6]. In Reddit, users publish multiple posts which are then commented by other users. To generate the corresponding post-connection graph, we regard the posts as nodes, and connect two posts with an edge if they are both commented or posted by the same *two* users, instead of only one user. If we don't make this restriction, all nodes commented or posted by one user would be fully connected, thus resulting in large cliques in the graph. We choose the data in January 2014 as the raw data and conduct the following preprocessing steps:

1. Delete the anonymous posts.
2. Sort the posts by their creation time and separate every 300,000 posts into a group.
3. For each group, we sort the subreddits by the total number of posts belonging to each subreddit. We exclude the subreddits which have either too many or too few posts. Then we choose the subreddits whose post number rank between 11 and 20 and remove the posts that don't belong to these subreddits.
4. Build a graph for each group based on the edge connection criterion.
5. Get the largest connected component in each graph.

The resulting graphs consist of 4017.6 nodes and 28697.6 edges on average. For the node feature, we concatenate each post's title and its description as the feature text. We use 300-dimensional GloVe CommonCrawl word vectors [7] to calculate the average word embedding in the text as the node features.

For transferable active learning on graphs from different domains, we use 5 benchmark datasets in addition to Reddit. Cora, Citeseer and Pubmed [20] contain citation networks of scientific publications, where each node represents a publication as a sparse bag-of-words feature vector, each edge corresponds to a citation link. Coauthor-Physics and Coauthor-CS [23] are co-authorship graphs, where the nodes represent authors and the edges indicate that two authors have co-authored a paper. Each node is represented by a bag-of-words vector of the keywords in the author's papers, while its label indicates the most active research field of the author.

The statistics of these dataset are shown in Table 5.

## C  Additional experimental results

### C.1  Transferable active learning on graphs across different domains

In this section, we report additional experimental results of transferable active learning on graphs across different domains. We follow the same experimental setting as in Section 4.3 and experiment on different training graphs. In Table 6, we show the results of training on Cora + Pubmed and testing on the remaining graphs. In Table 7, we show the results of training on Citeseer + Pubmed and

Table 6: Results of transferable active learning on graphs from different domains. Train on Cora + Pubmed, and test on the remaining graphs.

| Method | Metric | Citeseer | Reddit1 | Reddit2 | Reddit3 | Reddit4 | Reddit5 | Physics | CS |
|---|---|---|---|---|---|---|---|---|---|
| Random | Micro-$F_1$ | 59.74 | 81.64 | 91.47 | 87.63 | 85.26 | 86.26 | _84.81_ | 84.65 |
| | Macro-$F_1$ | 52.71 | 80.05 | 90.35 | 86.03 | 80.79 | 84.37 | 71.49 | 70.46 |
| AGE | Micro-$F_1$ | _65.60_ | _85.31_ | _92.40_ | _91.22_ | _86.90_ | _88.28_ | 83.93 | _86.69_ |
| | Macro-$F_1$ | **58.43** | _84.47_ | _91.00_ | _90.74_ | _83.10_ | _87.09_ | _75.28_ | _81.73_ |
| ANRMAB | Micro-$F_1$ | 62.87 | 83.14 | 88.55 | 85.95 | 81.51 | 83.58 | 82.06 | 86.48 |
| | Macro-$F_1$ | 55.90 | 82.21 | 86.12 | 84.03 | 74.56 | 79.40 | 70.92 | 78.61 |
| GPA (Ours) | Micro-$F_1$ | **65.76** | **88.14** | **95.14** | **92.08** | **91.05** | **90.38** | **87.14** | **88.15** |
| | Macro-$F_1$ | _57.52_ | **87.86** | **94.93** | **91.78** | **89.08** | **89.92** | **81.04** | **85.24** |

Table 7: Results of transferable active learning on graphs from different domains. Train on Citeseer + Pubmed, and test on the remaining graphs.

| Method | Metric | Cora | Reddit1 | Reddit2 | Reddit3 | Reddit4 | Reddit5 | Physics | CS |
|---|---|---|---|---|---|---|---|---|---|
| Random | Micro-$F_1$ | 66.85 | 81.64 | 91.47 | 87.63 | 85.26 | 86.26 | _84.81_ | 84.65 |
| | Macro-$F_1$ | 60.95 | 80.05 | 90.35 | 86.03 | 80.79 | 84.37 | 71.49 | 70.46 |
| AGE | Micro-$F_1$ | _70.08_ | _83.76_ | _92.56_ | _90.61_ | _86.94_ | _87.73_ | 84.68 | 86.33 |
| | Macro-$F_1$ | _66.94_ | _82.81_ | _91.61_ | _89.99_ | _83.15_ | _85.88_ | _77.25_ | _80.63_ |
| ANRMAB | Micro-$F_1$ | 68.50 | 83.14 | 88.55 | 85.95 | 81.51 | 83.58 | 82.06 | _86.48_ |
| | Macro-$F_1$ | 63.07 | 82.21 | 86.12 | 84.03 | 74.56 | 79.40 | 70.92 | 78.61 |
| GPA (Ours) | Micro-$F_1$ | **73.40** | **87.57** | **95.08** | **92.07** | **90.99** | **90.53** | **87.06** | **87.00** |
| | Macro-$F_1$ | **71.22** | **87.11** | **94.87** | **91.74** | **88.97** | **90.14** | **81.20** | **83.90** |

testing on the remaining graphs. We observe consistent trends with the results in Section 4.3 that our proposed method significantly outperforms the random selection baseline and the two active learning baselines. This suggests the effectiveness of our proposed method.

## C.2 Additional experiments on query budgets

In this section, we report additional experimental results on how the query budget influences the performance of the active learning policy.

First, we follow the experimental setting in Section 4.4 and report further results on other test graphs. Figure 4a shows the performance under different query budgets when using Cora as the test graph. We set the query budgets as $\{7, 14, 21, 35, 70\}$ respectively, since Cora has 7 classes. We observe that GPA achieves consistent performance gain over other baseline methods on different query budgets, which is consistent with the results in Section 4.4.

Second, we investigate how changing the training budget will influence the learned policy's performance on the test graph. Note that different training stage may require different kinds of labeled nodes, so the transfer may not be optimal when the test budget and training budget differ dramatically. Figure 4b shows the results on the Reddit dataset, where we use Reddit $\{1, 2\}$ for training and Reddit 4 for test. The x-axis corresponds to different query budgets on the training graphs, and each curve represents the performance on the test graph with a fixed query budget. We observe that a training budget of 30 is sufficient to yield good performance, and larger budgets will further yield more stable results with lower variance. When trained with large budgets, the learned policy is not optimal for small test budgets.

(a) Performance w.r.t. query budget on Cora

(b) Performace w.r.t. training budget

## Footnotes

[6]Reddit is an online forum where users create posts and comment on them. We use the January 2014 dump of Reddit posts downloaded from `https://bit.ly/3bumUtv` and `https://bit.ly/2Spg6G2`.

[7]`http://nlp.stanford.edu/data/wordvecs/glove.840B.300d.zip`