[Reviews · NeurIPS 2020]

Review 1

Summary and Contributions: The authors propose a graph policy network to deal with active learning problems on graphs. The query strategy is formalized as a Markov decision process, and a GNN-based policy network is learned with reinforce learning to select the most informative nodes so that the classifier could reach its best performance with the least labeling budget.

Strengths: The proposed graph policy network is trained with reinforce learning to take into account the long-term performance of active learning. From this point, the proposed method has advanced previous methods aimed to only satisfy the short-term criterion at each step. Node interactions are also captured in GNN-based policy network to better measure node informativeness in local neighborhood. The effectiveness of the proposed method is evaluated on transferable active learning on graphs from the same domain and across different domains.

Weaknesses: The proposed framework is proposed to reduce the labeling budge for transferable active learning problems on graphs. However, experiments have issues with clarity. - In Section 4.2 & 4.3, It is unclear how many labeled nodes are used to train the policy network? - Why ANRMAB performs much worse than AGE? In theory, ANRMAB is expected to outperform AGE as it uses multi-armed bandit to adjust the weights of three heuristics. - Given this paper is proposed to address transferable active learning, the focus of experiments should be on measuring classification performance w.r.t. different query budgets, rather than overall performance in Tables 1&2. - In Section 4.4, it is hard to see the effect of labeling budgets for training the policy network and more importantly the effect of query budgets on classification performance. This part of experiments could be better designed to support the claims made.

Correctness: Most of the claims and method are correct, but empirical evaluation could be better set up to support the claims made.

Clarity: In general, this paper is well written. The methodology is clearly presented, making it relatively easy to follow.

Relation to Prior Work: Yes, this work has done a good job in reviewing the previous work.

Reproducibility: Yes

Additional Feedback: I have read the author rebuttal. It would be good if the authors could explain how the model is transferred to the test graph more explicitly.


Review 2

Summary and Contributions: The paper studies active learning for graphs using reinforcement learning.

Strengths: This is an interesting paper that aims at reducing the label annotation for GNNs. The proposed algorithm seems sound and results are good.

Weaknesses: How the improvement of the identified limitation "Ignoring Long-term Performance" in page 1 was tested in the numerical experiments? It was not clear to me what are the definitions of the reward R and the measure M in Eq.1 and Eq.4?

Correctness: Likely.

Clarity: The paper is nicely written.

Relation to Prior Work: Comparisons with previous graph active learning works [8,13,12,5] could be better explained. What are the new advantages of the newly proposed technique?

Reproducibility: No

Additional Feedback:


Review 3

Summary and Contributions: This submission proposed a graph policy network for zero-shot transferrable active learning to query node labels for semi-supervised node classification on a "target" graph by training the policy network using multiple labeled "source" graphs. Experimental results based on graphs from the same or different domains demonstrate the efficacy of the proposed method. Updated during rebuttal: I truly appreciate the authors's extra efforts after rebuttal for clarification. As it is critical to clearly present the transferrable active learning with the policy network, I would strongly suggest that the authors shall rewrite Section 3 to better present all the transferrable active learning components, especially clearly stating that the learned policy network is applied to node-based representations.

Strengths: 1. Collected training data with labels in general can be difficult and costly. The paper provides a transferrable active learning strategy for semi-supervised node classification with graphs, which can potentially address this challenge in this specific setup.

Weaknesses: 1. The paper focused no node classification performance evaluation. There are more tasks with graphs, including link prediction mentioned by the authors. It would be interesting to check the performances of the proposed active learning strategy , for these tasks. 2. More critically, the presentation needs to be significantly improved. Specifically, it is not clear how graph policy network and the one for node classification interact with each other. It is not clear either how the learned policy network can be applied to the target graph if source and target graphs have different topology or target and source have different number of classes. Updated during rebuttal: I truly appreciate the authors's extra efforts after rebuttal for clarification. As it is critical to clearly present the transferrable active learning with the policy network, I would strongly suggest that the authors shall rewrite Section 3 to better present all the transferrable active learning components, especially clearly stating that the learned policy network is applied to node-based representations.

Correctness: The idea of training a graph policy network does make sense for active learning in graph neural networks. However, the clarity is lacking.

Clarity: In addition to the aforementioned presentation problems, there are numerous other problems: 1) For node class prediction probabilities, there is no clear description how they are computed. 2) In equation (2), throughout the paper, it is not clear which evaluate metric was used to estimate reward. Is it micro-F1, macro-F1, directly or something else? 3) The definition in equation (3) is not clear as the authors did not even definite the action space. Is that defined for each node or the whole graph? The paper definitely needs careful rewriting.

Relation to Prior Work: The authors implemented the deep reinforcement learning for active learning strategy in [7] for the graph node classification problem. There are also existing active learning methods for graph data, including the ones [3], [5], and [10] as the authors discussed in the related work. However, the performance comparison with [5] was not reported.

Reproducibility: Yes

Additional Feedback: Update during rebuttal: The authors have clarified many places during the rebuttal, which is highly appreciated. I do hope they can rewrite the Section 3 (with updated Figure 1) and Appendix significantly to make the proposed method presentation more self-explaining.


Review 4

Summary and Contributions: This paper presents a novel active learning strategy for GNNs which leverages a GNN-based policy network. Experiments show that in a semi-supervised node classification setting, the proposed approach can greatly reduce the label budget.

Strengths: - GPA is an elegant approach to address the widespread scarcity of labels in graph ML tasks - the experiments are very convincing, both in terms of raw performance, and in terms of the ablation study, query budgets, etc. After reading the paper, I was left with the impression that each design choice made in GPA has been carefully vetted by the authors

Weaknesses: - the 5 different graphs from "different domains" are in fact covering the same domain (i.e., they are all citation networks) - Figure 2.Left would have greatly benefited from an additional curve obtained from the performance of a recent, SotA GNN architecture - lack of discussion on the training overhead induced by GPA

Correctness: To the best of my knowledge, the proposed method is sound, The empirical methodology is exhaustive and robust. I inspected the code as well, and it appears to be of good quality.

Clarity: Yes, the paper is well written and well structured. It was honestly a pleasure to read.

Relation to Prior Work: The discussion on the related work is reasonably extensive. I would have liked to see more recent GNN architectures, both in the related work and in the experiments (to find out if GPA would carry the same benefits).

Reproducibility: Yes

Additional Feedback: I thoroughly enjoyed reading this paper, and I found the GNN-based policy network to be a very elegant and effective idea. I raised a few concerns in the previous sections of my review, which I'm confident the authors will be able to address in the next revision of the manuscript. As an additional minor comment, Figure 1 could be made more readable -- for instance, it is hard to understand that the 4 graphs in the policy network \pi panel represent different feature aggregations. === Response === I would like to thank the authors for their thorough rebuttal, and for their quick responses. The additional details provided by the authors confirmed my understanding of the paper, and solidified my very positive score.

[Author Response · NeurIPS 2020]

We thank all the reviewers for their constructive feedback. Below are the responses to each reviewer.

**Reviewer #1: (1) Number of labeled nodes to train the policy network.** We use all the labeled nodes in the training graphs, as one could very easily find some fully labeled graphs to train the policy network. **(2) About ANRMAB.** Yes, theoretically you are right. However, to learn good weights of different heuristics for ANRMAB, at least a moderate number of labeled data are required. In our setting, we focus on very limited query budget, with which it is very difficult to learn good weights. **(3) Performance w.r.t. query budgets.** We agree that it is important to report the classification performance w.r.t. different query budgets in active learn-ing. As an example, we have illustrated the corresponding curves on Reddit 4 in

Fig. 1: Performance w.r.t query budget on Cora

Section 4.4 (Paper). Here, we provide additional results in Fig. 1, where GPA is trained on Reddit $\{1, 2\}$ and evaluated on Cora. We observe similar trends to the results in Section 4.4 (Paper). **(4) Concerns on Table 1&2.** The purpose of Table 1&2 is to compare the performance of different active learning algorithms under the same query budgets of $(5\times\#\text{classes})$. We have compared classification performance w.r.t. different query budgets in Section 4.4 (Paper) and Fig. 1. **(5) Concerns on Section 4.4.** This is a very good point! Following your suggestion, we fix the test budget and change the training budget to see how the performance varies. Fig. 2 shows the results on the Reddit dataset, where graph 1&2 are used for training and graph 4 for testing. The x-axis of the figure corresponds to different query budgets on the training graphs. The results show that a training budget of 30 queries is sufficient to yield good performance, and more budgets will further yield more stable results with lower variance. For the effect of query budgets on classification performance, it has been discussed in the aforementioned answer (3). We will add more results on different graphs in the revised version.

**Reviewer #2: (1) About "Ignoring Long-term Performance".** All the baseline methods except ANRMAB greedily choose the node with the maximal surrogate criterion score to label, which ignore the long-term performance. In contrast, our method uses reinforcement learning to label nodes with maximal long-term performance gain. In experiment, our method outperforms all the greedy methods, which proves our claim. **(2) Definition of the reward and evaluation metric.** Empirically, we use Micro-$F_1$ score of the classification GNN on test sets as the evaluation metric $\mathcal{M}$ to generate the reward signal $R$. **(3) Relation to prior work.** Most previous methods use different kinds of greedy strategies to identify informative nodes to label. In this paper we formulate the problem of active

Fig. 2: Performace w.r.t. training budget

learning on graphs as a sequential decision process and propose to train an active learning policy network to maximize the long-term performance score on the end task. We will discuss this in more details in the revised version.

**Reviewer #3: (1) Experiments on other tasks.** We agree that it would be interesting to evaluate the proposed algorithm GPA on other tasks. Indeed, GPA is very general and can be easily applied to different tasks by changing the reward functions accordingly. Here we take node classification as an example, which is the most fundamental problem on graphs. **(2) Questions about zero-shot node classification.** This is a misunderstanding. Our paper actually focuses on "zero-shot transfer learning" instead of "zero-shot node classification". "Zero-shot transfer learning" means that the *policy network* learned on labeled training graphs can be directly applied to the unlabeled test graphs without any further fine-tuning. In addition, the active learning setting mainly focuses on identifying informative nodes to label for supervised learning, which is different from unsupervised learning setting. **(3) Interaction between graph policy network and the one for node classification.** The graph policy network selects unlabeled nodes for annotation to train the node classification network. Meanwhile, the performance of the classification network is used as rewards to train the graph policy network. **(4) Transferring to other graphs.** This is a good point. Indeed, our algorithm is not sensitive to the number of classes between source and target graphs, because all the considered state features are not sensitive to the number of classes. Our algorithm parameterizes policy networks with GNNs, which naturally generalize to graphs with different topology. In experiment, we evaluate GPA on graphs with different numbers of classes and different topology, and show compelling results. **(5) Indices of state features.** The fourth state feature is defined as $\mathbf{s}_v^t(4)$ in the equation following L115 on page 3. **(6) Node class prediction probabilities.** Following existing literature, we apply a linear softmax classifier on top of the node representations learned by the classification GNN to get the node class probability. **(7) Evaluation metric for reward.** Empirically, we use Micro-$F_1$ as the evaluation metric for reward generation. **(8)Action space.** Remember that the action of the policy network is to select an unlabeled node and query for its label in each query step, and thus the action space is defined as the unlabeled nodes in the training set. **(9) Comparison with [5].** [5] considers batch-mode active learning on heterogeneous graphs, which cannot be directly applied to our setting. Also, the idea of [5] is very similar to ANRMAB, where the problem is both formulated with multi-armed bandit, and thus we mainly compare against ANRMAB in the paper.

**Reviewer #4:** We appreciate your positive feedback, and will revise the paper according to your suggestions.

[Meta-Review · NeurIPS 2020]

The paper proposes a new active learning method that learns on training graphs a graph policy network, which is applied to query the labeled nodes of a test graph. Initially, there were serious concerns about the clarity of technical details, in particular, how "zero-shot transfer learning" on node classification was defined and realized. The AC shared similar concerns and had reached to the authors via CMT to ask for additional response. The additional response from the authors, which described in detail the techincal details on how the policy network was applied to a test graph, had helped address these concerns. Consequently, a general consensus of the reviewers was reached on accepting the paper (R1 commented during the discussion: "I agree that maybe more explanations about the transferability of the model could have been elaborated in the main body of the paper, but I do think this is an interesting paper and its overall quality is good. So I am leaning towards a weak accept"; R3 commented to the AC: "I raised the score from 3 to 5, just to be fair to other submissions, which may not have the opportunity to be discussed seriously with another chance to further clarify their papers."). The AC encourages the authors to incorporate their additional response sent via CMT to their final revision, to clearly enhance the clarity of the technical details, in particular, how the graph policy network is applied to query labels on a test graph.